# Links between Inflammation and Postoperative Cancer Recurrence

**DOI:** 10.3390/jcm10020228

**Published:** 2021-01-10

**Authors:** Tomonari Kinoshita, Taichiro Goto

**Affiliations:** 1Division of Thoracic Surgery, Japanese Red Cross Ashikaga Hospital, Tochigi 326-0843, Japan; kinotomo0415@gmail.com; 2Lung Cancer and Respiratory Disease Center, Yamanashi Central Hospital, Yamanashi 400-8506, Japan

**Keywords:** cancer, cancer recurrence, clinical index, inflammation, postoperative recurrence, resection

## Abstract

Despite complete resection, cancer recurrence frequently occurs in clinical practice. This indicates that cancer cells had already metastasized from their organ of origin at the time of resection or had circulated throughout the body via the lymphatic and vascular systems. To obtain this potential for metastasis, cancer cells must undergo essential and intrinsic processes that are supported by the tumor microenvironment. Cancer-associated inflammation may be engaged in cancer development, progression, and metastasis. Despite numerous reports detailing the interplays between cancer and its microenvironment via the inflammatory network, the status of cancer-associated inflammation remains difficult to recognize in clinical settings. In the current paper, we reviewed clinical reports on the relevance between inflammation and cancer recurrence after surgical resection, focusing on inflammatory indicators and cancer recurrence predictors according to cancer type and clinical indicators.

## 1. Introduction

Cancer recurrence remains one of the greatest concerns for patients with cancer [1] and markedly contributes to the poor prognosis and quality of life associated with several cancer types [2]. Many factors, including remnant cancer cells at the tumor margin after surgery and intrinsic cellular resistance to treatment, contribute to cancer recurrence [3,4]. Resistance to anticancer medications occurs at the tissue level due to the modification of the cancer microenvironment by cancer-promoting mediators [5,6]. Several resistance mechanisms associated with the cancer microenvironment are well recognized [7,8], including the genetic modulation of cancer cells that results in the expression of tumor-promoting metabolic factors that induce invasion, thereby promoting resistance [9,10].

The inflammatory microenvironment is a crucial component in most tumors, considering the important role that the inflammatory response plays in tumor development and metastasis. The potential for tumor metastasis depends on complex and dynamic interactions between cancer cells, inflammatory cells, immune cells, and stromal elements within the tissue from which they originate. The phases of metastasis can be classified into four distinct stages. The first stage involves the epithelial-mesenchymal transition, during which, cancer cells acquire fibroblast-like properties that allow them to promote their motility, invade the epithelial endothelium/basement membrane, and reach exudative blood and lymphatic vessels [11]. The second stage involves cancer cell invasion of the blood and lymphatic vessels, which may be facilitated by inflammation via the production of cytokines that upregulate vascular permeability. In the third stage, cells have begun to metastasize, survive, and migrate via the circulatory system [12]. In the final stage, single metastatic progenitor cells interact with stromal, inflammatory, and immune cells and begin to proliferate [13]. Cancer cell invasion necessitates protease degradation of the extracellular matrix around the invasive forefront, with inflammatory cells being a critical source of extracellular matrix-degrading proteases. Moreover, Th2 cytokines stimulate matrix metalloproteinase expression, invasive potential, and metastasis, whereas other cytokines that constitute the Th1 response inhibit tumor progression [14]. Thus, the complicated association between cancer cells, immune function, and inflammatory cells regulates tumor progression and the establishment of distant metastasis [15,16,17,18,19].

Over the past two decades, researchers have identified several molecules that have an important role in inflammation, including vascular endothelial growth factor (VEGF), tumor necrosis factor, chemokines, cyclooxygenase (COX)-2, interleukin-1 (IL-1), interleukin-6 (IL-6), 5-lipoxygenase, matrix metalloproteases, and cell surface adhesion molecules. However, it remains unclear whether acute and chronic inflammatory conditions or different types of acute inflammation produce similar results. Moreover, studies are yet to elucidate whether the inflammatory burden of stored blood product transfusion is similar to that of acute infectious stimuli. As such, it remains to be determined whether an acute inflammatory burden (e.g., perioperative transfusion) in an immunosuppressive environment (e.g., perioperative stress) complicates the patient’s condition and creates a pro-tumor environment that facilitates distant metastases. To explore this, the present paper reviewed numerous clinical reports examining the potential association between inflammation and cancer recurrence after surgical resection, focusing on inflammatory indicators and cancer recurrence predictors in major cancer types and their clinical indicators.

## 2. Types of Cancer

### 2.1. Lung Cancer

Lung cancer remains the top cause of cancer-related death in the world [20]. Although surgery with curative intent is the treatment of choice for patients with early-stage cancers, postoperative complications frequently occur in the early phase after the surgery [21,22]. Moreover, patients with lung cancer often develop postoperative recurrence in the late phase after the surgery [23,24,25]. Although both acute postoperative complications and cancer recurrence in the late phase are regarded as major problems after the surgery, the interplay between postoperative complications and recurrence in the late phase among patients who have undergone lung cancer surgery is an issue that has only recently been investigated [26,27,28,29,30].

According to previous clinical studies, it was shown that the inflammatory response has a critical function in tumor invasion, progression, and metastasis by promoting tumor angiogenesis and decreasing anticancer effects through the upregulation of inflammation [31,32]. Recently, inflammatory biomarkers, such as C-reactive protein (CRP), the neutrophil-to-lymphocyte ratio (NLR), the monocyte-to-lymphocyte ratio (MLR), and the platelet-to-lymphocyte ratio (PLR) were found to be associated with cancer prognosis [33]. For instance, Mizuguchi et al. examined the prognostic value of the NLR in patients with completely resected stage I non-small cell lung cancer (NSCLC), whereas Nojiri et al. showed that patients with postoperative respiratory complications had markedly higher postoperative white blood cell counts, CRP levels, and cancer recurrence rates than those without [34]. Meaney et al. reported that among patients with stage I lung adenocarcinoma, those with high IL-6 and IL-17A levels had a lower 5-year survival rate (46%) than those with low levels of both markers (93%), with a similar trend having been observed for the prognostic signatures of IL-6 and IL-17A in an independent data set [35]. Multivariate analyses in another study revealed that although the NLR was not predictive of overall survival (OS), a high NLR was an independent risk factor for recurrence, in addition to age, stage, tumor differentiation, and lymphatic invasion. The same study concluded that a lower preoperative NLR was correlated with a better prognosis and a lower systemic inflammatory status in patients with stage I NSCLC [36].

Additionally, the systemic immune-inflammation index (SII), a new inflammatory biomarker that is defined as (platelet count × neutrophil count)/(lymphocyte count) [37], has been found to potentially have a high prognostic value for patients with cancer [38]. Indeed, a meta-analysis of the SII showed that a higher pretreatment SII indicated markedly poorer OS, disease-free survival (DFS), and progression-free survival (PFS) with high heterogeneity, as well as cancer-specific survival (CSS). Given that the SII has a more accurate prognostic value in NSCLC than the NLR and the PLR, the pretreatment SII may be a useful prognostic marker for NSCLC and may be more helpful for prognostic assessment and treatment strategy development [39]. The advanced lung cancer inflammation index (ALI) has also been reported to be associated with survival in patients with lung cancer. In fact, Zhang et al. showed that among NSCLC cases, those with a low ALI had poorer OS than those with a high ALI. Subgroup analyses further revealed that a low ALI had a significant negative prognostic value in NSCLC, with a low ALI being clearly associated with lower PFS and recurrence-free survival (RFS) in patients with NSCLC. Based on these results, the aforementioned study concluded that the pretreatment ALI was indicative of poor prognosis among patients with NSCLC, suggesting its potential utility as a tumor marker for survival prediction among the same patients [40]. The Glasgow Prognostic Score (GPS), which includes albumin and CRP levels, can predict postoperative survival in stage I and advanced NSCLC patients [41,42,43]. Therefore, these clinical indices are reliable predictive markers for post-treatment survival in lung cancer.

Another interesting study investigated the relationship between cancer recurrence and lung scars. Inflammation-mediated events that can occur independently of inflammation, such as reactive oxygen species production, growth factor activation (for wound repair), and altered signaling processes that activate cell proliferation (replacing necrotic/apoptotic tissue cells), have all been considered risk factors for various cancers. Taking lung scar cancer as an example, this study discussed the mechanisms of inflammation that is associated with recurrent *Mycobacterium tuberculosis* infection in the context that they may be the primary or exclusive cause of cancer. The production of reactive oxygen species, cytokines, leukotrienes, and prostaglandins in lung tissues is greatly upregulated by a cell-mediated immune response to *M.tuberculosis*-infected macrophages. These responses result in extensive fibrosis that is associated with recurrent infection, possibly causing the reduced clearance of lymphatic and lymph-related particles from the site of infection. Moreover, the inhibition of p21 synthesis increases the rate of cell division and accelerates cell cycle progression from G0 arrest to the G1 phase, the G1 phase to the S phase, and the G2 phase to the M phase. Furthermore, increased oxidative DNA damage and the suppression of apoptosis due to increased BCL-2 synthesis, coupled with enhanced progenitor cell mutagenesis and increased angiogenesis due to COX-2 products, create an environment that is conducive to tumorigenesis. The provided evidence appears to suggest that some scar cancer cases would not have existed without an inflammatory response to recurrent infections. Appropriate lifestyle and dietary changes would provide various anti-inflammatory effects and reduce the risk of inflammation-induced cancer [44].

### 2.2. Breast Cancer

Molecular mechanisms associated with inflammation have been considered to play an important role in the development of breast cancer, which is the leading cause of death among women. Studies have shown that proinflammatory cytokines alter tumor cell biology and promote cancer progression and metastasis by activating stromal cells in the tumor microenvironment, including fibroblasts, vascular endothelial cells, and tumor-associated macrophages in breast cancer [45,46,47,48]. Systemic inflammation may also alter the status of the vasculature in such a way as to promote micrometastasis leaching, transplantation, and growth [45,48], or reactivate dormant tumors at distant sites [49]. The new significance of inflammation in breast cancer progression is noteworthy considering that primary breast tumors are rarely accompanied by significant inflammation per se [50,51]. However, biological processes that induce metastasis or sustain residual disease during treatment may be totally different from those that promote primary tumor formation [52]. According to Paget’s analogy [52], chronic inflammation fertilizes the soil of systemic tissues and promotes the growth and dissemination of metastatic seeds. Analyses comparing molecular characteristics and the location of primary and recurrent tumors will elucidate the extent to which inflammation contributes to the regrowth of the primary tumor, disease recurrence, development of micrometastases, or the emergence of entirely new malignancies. Oshi et al. demonstrated that intratumoral angiogenesis is associated with inflammation, immune reactions, and metastatic recurrence in breast cancer [53]. Although the direct correlation between inflammation in the primary tumor and cancer recurrence was not revealed, a relationship via angiogenesis may exist and have an important role in tumor recurrence.

Chronic inflammation associated with obesity is now recognized as an important condition that promotes carcinogenesis and cancer progression in patients with breast cancer, primarily in postmenopausal women with tumors expressing estrogen and progesterone receptors. Altered levels of several inflammatory mediators that regulate aromatase and estrogen expression are among the mechanisms that increase the risk of breast cancer in patients with obesity. The role of local adipose inflammation and macrophages as determinants of the risk of breast cancer recurrence and prognosis has also received increasing attention. The obesity–inflammation axis provides different molecular signaling pathways for potential pharmacological targets and therapeutic interventions. Moreover, the increasing rates of obesity worldwide have prompted further investigation into recent findings linking inflammation and breast cancer [54]. With regard to clinical indicators, dietary inflammatory indexes (DIIs) have been associated with breast cancer incidence, as well as survival and recurrence [55].

The following discussion details the current prospects for alleviating the impact of systemic inflammation on the recurrence of breast cancer. Polymorphisms of the cytokine gene are reported to have some effect on breast cancer progression, which also suggests that even a partial reduction in inflammatory signaling may be protective when maintained over time [56,57]. Although long-term nonsteroidal anti-inflammatory drug use has been associated with a lowered risk of primary breast cancer [58,59], its efficacy as adjuvant therapy after the successful treatment of early-stage disease remains yet to be verified. Although tamoxifen clearly reduces acute phase protein (APP) levels [60,61,62], part of the protective effect of endocrine therapy may be attributed to its anti-inflammatory effects. The long-term use of other anti-inflammatory agents, such as COX2 inhibitors, cytokine antagonists, and glucocorticoids, has been associated with side effects that could restrict their possible role in an adjuvant setting. The most beneficial approach may be to target upstream factors that promote chronic inflammation, such as inactivity and adiposity [63,64]. In an analysis adjusting for age, physical activity, and adiposity, Pierce et al. found that residual variation in APP levels was predictive of breast cancer recurrence. While this does not deny the significance of adiposity and physical activity, it strongly suggests that other factors influencing chronic inflammation, such as smoking, subclinical infections, major depression, low socioeconomic status, and heavy drinking of alcohol, affect the risk of breast cancer recurrence [63,64,65,66,67]. Relieving the impact of such lifestyle-related factors is a difficult task for both patients and clinicians but one that many breast cancer survivors need to address if they are to understand the potential for preventing breast cancer recurrence and the development of other cancers and cardiovascular diseases, as observed by Pierce et al. According to the presented data, disease recurrence was markedly promoted in only the top third of the APP distribution, suggesting that resource-intensive lifestyle interventions may target some patients based on the inflammatory biomarkers of disease risk. Despite the specific treatment, the findings highlight the need to address the broader environment of patients’ behavior and global health, given their effects on localized neoplastic disease and clinically occult breast cancer recurrence. Taking a systemic approach to controlling minimally remnant disease may identify new possibilities for reducing the risk of recurrence after the successful treatment of early-stage breast cancer [68].

### 2.3. Esophageal Cancer

Although esophageal cancer occurs less frequently than other cancers, it is characterized by a wide range of lesions involving the neck, chest, and abdomen, and is difficult to treat [69]. While various methods for treating esophageal cancer currently exist, surgery is still the treatment of choice. The principles of surgical treatment include reducing surgical trauma, promoting patient recovery, and increasing treatment safety, which are based on the premise of complete tumor resection and lymph node dissection [70]. Given that patients with postoperative recurrence or metastasis lose the opportunity for surgical treatment, conservative interventions, such as chemotherapy, radiotherapy, and biotherapy, may be the most effective approach in such cases [71]. Studies suggest that approximately 40% of patients with esophageal cancer who undergo curative surgery develop postoperative recurrence/metastasis within 2 years [72] and there is still no standard treatment regimen for the postoperative recurrence/metastasis of esophageal cancer. With the recent changes in people’s lifestyles, the incidence of esophageal cancer has shown a clear increasing trend in younger people [73]. After examining the correlation between recurrence and insulin resistance, glycolipid metabolism, stress, inflammation, and serum p53 concentration after radical surgery in 80 patients with esophageal cancer who underwent surgery, Zheng et al. concluded that complex hypertension, hyperlipidemia, and diabetes mellitus were independent risk factors for postoperative recurrence of esophageal cancer. Moreover, they found that earlier postoperative recurrence was associated with higher fasting blood glucose, insulin resistance index, and total cholesterol levels, as well as stronger inflammatory and oxidative stress responses. This is one of only a few papers that showed an association between high inflammatory responses and high recurrence rates in the perioperative period [74]. Lindenmann et al. showed that the GPS score was significantly associated with survival and was considered to be a poor prognostic marker in advanced esophageal carcinoma [75]. In general, most patients complained about spontaneous weight loss at the first visit. It frequently happens that patients with esophageal cancer, as well as gastric cancer, lose weight continuously, resulting in a compromised functional performance status during treatment. Thus, we should pay attention to such confounding factors, which can influence not only treatment but also post-treatment survival when we use these clinical indexes in esophageal cancer.

### 2.4. Gastric Cancer

Gastric cancer is the fifth most common cancer and the third leading cause of mortality in the world [76]. Currently, surgery is the treatment of choice for early-stage gastric cancer. Even after curative resection, recurrence or metastasis occur in approximately 35–70% of patients within 5 years [77]. During the past few decades, various adjuvant chemotherapy regimens have been tested with the hope of controlling postoperative recurrences. Accordingly, patients with advanced gastric cancer have been found to benefit from fluoropyrimidine-based chemotherapy [78], whereas those with locally advanced gastric cancer were able to achieve an improved OS with the administration of capecitabine and platinum-based regimens [79]. Moreover, patients whose tumors have recurred or metastasized may have a better prognosis when chemotherapy is combined with targeted therapy [78]. However, approximately 50% of all gastric cancers are not responsive to treatment, and only a small number of patients achieve a stable disease status or a partial response to treatment [80].

As mentioned previously, lymphocytes play an important role in cancer immune surveillance and defense via the induction of cytotoxic cell death and the inhibition of tumor cell proliferation and migration [45], thereby modulating the host’s immune response to malignancy [81]. Some interleukins (IL-1β/IL-4/IL-6/IL-13) may correlate with the recurrence of gastric cancer [82]. Several immune- and inflammation-based prognostic scores, such as MLR and NLR, have been developed as predictors of survival and recurrence across various cancer types, including gastric cancer [83,84,85]. However, integrated indices based on peripheral lymphocyte, neutrophil, and monocyte counts that may better reflect the balance between the host’s immune and inflammatory statuses have not yet been reported for gastric cancer. In addition, the effects of peripheral lymphocyte, monocyte, and neutrophil counts on gastric cancer recurrence and metastasis remain to be studied in detail.

### 2.5. Hepatocellular Carcinoma

Hepatocellular carcinoma (HCC) is the sixth most common cancer and the fourth leading cause of cancer mortality worldwide [20]. According to the Scientific Registry of Transplant Recipients, there has been a steady increase in the number of liver transplants, with an annual rise of 3% as of 2017. The same report revealed that among patients waiting for liver transplants, the number of patients with HCC has increased, accounting for nearly 10% of transplants [86]. HCC recurrence can be attributed to its strong correlation with cirrhosis and the highly vascular nature of the liver. This lowers the long-term prognosis of localized treatments, such as ablative therapy and resection; therefore, a liver transplant remains the mainstay for the curative treatment of HCC [87].

Several studies have tried to identify HCC recurrence predictors before transplantation to enhance patient selection and transplant outcomes. The development of cancer produces a chronic inflammatory response that causes neovascularization and disrupts normal immune pathways [46]. The involvement of platelets and leukocytes, along with the recruitment of growth factors and interleukins, mediates tumor growth [88,89]. The disruption of normal homeostasis in these pathways can be detected using straightforward, easily measured markers that may represent cancer aggressiveness and survival. Inflammatory marker ratios derived from peripheral blood count testing have been applicable to various cancers and chronic inflammatory conditions [90,91,92]. Meanwhile, others have previously shown that NLR can be used to predict relapse in HCC [93]. Ismael et al. assessed inflammatory markers for HCC based on the total blood count to predict the OS and RFS in patients after liver transplantation. Between 2001 and 2017, 160 patients with HCC who were eligible for liver transplantation were retrospectively reviewed, among whom, 74.4% had hepatitis C virus as the background factor of HCC. Accordingly, although patients with an NLR of ≥5, a derived NLR (dNLR) of ≥3, and a low lymphocyte-to-monocyte ratio (LMR) (<3.45) had markedly worse model of end-stage liver disease (MELD) scores, no correlation was observed between the MELD score and those with a high PLR (>150). Among the cancer characteristics, low LMR was in close association with tumor presence and microvascular infiltration at the time of transplantation. Moreover, a lower NLR and dNLR and a higher LMR tended to be associated with a better OS, a higher LMR tended to be associated with a higher RFS, and a low PLR was associated with significant OS and RFS. Thus, while previous studies on HCC identified NLR as a biomarker for tumor burden and survival, Ismael et al. emphasized that PLR was a superior surrogate for identifying complications and RFS in HCC transplant recipients [94].

### 2.6. Bile Duct Cancer

Inflammation-based prognostic scores have also been associated with cancer survival and recurrence in hepatobiliary and pancreatic cancers. Fujiwara et al. used univariate analysis to evaluate clinicopathological factors, including various prognostic scores, DFS, and OS, in 121 patients with distal extrahepatic cholangiocarcinoma who underwent a pancreaticoduodenectomy between 2000 and 2015. Accordingly, they showed that the GPS, which considers a combination of albumin and CRP levels, was an independent risk factor for poor prognosis and cancer recurrence. As such, they concluded that the GPS score is an independent marker for cancer recurrence in patients with distal extrahepatic cholangiocarcinoma and is superior to other prognostic scores [95].

### 2.7. Colorectal Cancer

Although radical mesorectal resection and neoadjuvant therapy have improved the prognosis of patients with rectal cancer, locally recurrent rectal cancer after a complete resection still occurs in <5–10% of patients [96,97,98]. However, surgery remains the only option for improving symptom control, quality of life, and long-term survival among those with locally recurrent rectal cancer. R0 resection is an important prognostic factor for improving long-term survival in locally recurrent rectal cancer [99]. However, given that radical surgery is highly invasive, complications occur at a high rate [100]. Previous studies have demonstrated that inflammation is associated with malignant cell proliferation and survival [45,101]. However, it remains elusive whether intra-abdominal and neck inflammation after radical surgery for locally recurrent rectal cancer affects prognosis. After analyzing prognostic factors associated with OS and RFS in patients undergoing radical surgery for locally recurrent rectal cancer, Johnson et al. identified massive perioperative bleeding as the only factor associated with the development of intra-abdominal/intraspinal inflammation. Moreover, they showed that intra-abdominal/intraspinal inflammation after curative surgery for locally recurrent rectal cancer was associated with a poor prognosis [102]. Another study by Varkaris et al. demonstrated that a circulating inflammation signature is a strong prognostic factor for RFS and OS in patients with metastatic colorectal cancer [103].

Conversely, anemia itself has been found to be a poor prognostic factor in patients with colorectal cancer. Conventionally, anemia in colorectal cancer has been supposed to be related to iron deficiency due to frank or latent gastrointestinal bleeding and the associated microcytosis [104]. Therefore, considerable efforts have been made toward iron replacement therapy during the perioperative period. However, a recent observational study reported that normocytic anemia was approximately twice as prevalent as microcytic anemia in patients with colorectal cancer who underwent surgery for curative purposes [105]. Meanwhile, although normocytic anemia was associated with a poor prognosis, no such association was observed with microcytic anemia. Notably, normocytic anemia was significantly associated with preoperative systemic inflammation, which was characterized by a modified GPS [106]. Moreover, iron replacement therapy was not likely to be beneficial in this patient group. Unfortunately, no such data was obtained and the size of the cohort did not allow for meaningful subgroup analysis. McSorley et al. reported on the prevalence and prognostic impact of normocytic anemia in patients receiving radical treatment for colorectal cancer. They found that normal cellular anemia was associated with systemic inflammation and poor CSS. Considering that inflammation may cause both disease recurrence and anemia in such patients, focusing on and treating this process may improve both [107].

### 2.8. Uterine Carcinoma

Cervical cancer is the second most common type of cancer among women in countries without access to screening and prevention programs. Depending on the initial tumor stage, 8–61% of women with cervical cancer have recurrence, typically within 2 years following treatment completion [108]. After evaluating the role of GPS in patients with cervical cancer after radical resection, Seebacher’s group found that a high GPS assessed before the initial treatment was independently associated with a shorter survival time [109,110,111]. They also reported that a higher GPS upon relapse, a history of radiation therapy, and the presence of peritoneal carcinomatosis or multiple sites of relapse were independently associated with shorter post-relapse survival in patients with recurrent cervical cancer. Unfortunately, only a few papers have shown an association between a high inflammatory response and high recurrence rates among patients with uterine carcinoma.

### 2.9. Prostate, Urothelial and Bladder Cancer

Prostate cancer is the most common type of malignancy and the leading cause of cancer death in men in the world [112]. Prostate cancer usually has a long natural history from diagnosis till death caused by cancer progression. Similar to breast cancer, inflammation is the primary etiology of prostate cancer development. Acute or chronic inflammation can lead to carcinogenesis, as well as prostate cancer progression [113,114,115,116]. Chronic systemic inflammation, which is associated with the risk of several cancer types, including breast, colon, and pancreatic cancer, has been attributed to lifestyle [117]. Hayashi et al. found that medications and diets that modulate the immune status or suppress the inflammatory response, such as metformin, aspirin, nonsteroidal anti-inflammatory drugs, statins, and soy isoflavones, may be clinically beneficial for patients with prostate cancer [118].

Regarding the postoperative recurrence of prostate cancer, Irani et al. evaluated the prognostic value of prostatic stromal inflammation in surgically treated and localized prostate carcinoma [119]. Notably, they reported that the inflammation grade in malignant tissue had a stronger impact on RFS than preoperative serum prostate-specific antigens and the pathologic stage. Soria et al. retrospectively examined the prognostic role of the modified GPS (mGPS) for the prediction of oncological outcomes in a large multicenter cohort of patients with upper tract urothelial carcinoma who underwent radical nephroureterectomy. Accordingly, they found that mGPS was independently associated with clinicopathological features and prognostic outcomes after a radical nephroureterectomy [120]. Furthermore, after investigating the clinical impact of postoperative (3 months after radical cystectomy) inflammatory biomarkers, including NLR and the hemoglobin-to-platelet ratio, which were strongly associated with RFS, CSS, and OS, Albisinni et al. found that the preoperative and postoperative inflammatory status of ureter-related cancers may affect cancer recurrence.

## 3. Inflammation-Related Conditions

### 3.1. Necrosis

Anticancer therapy often induces tissue hypoxia and hypometabolism, causing necrotic cell death, which in turn triggers an immune response through mediators secreted by necrotic cells [121,122]. The disruption of cell membranes after necrosis leads to the secretion of intracellular signals known as damage-associated molecular patterns [123], which function as endogenous danger signals that promote the inflammatory response in aseptic inflammation [124]. Damage-associated molecular patterns lead to tissue repair and wound healing, as well as promoting cancer invasion in chronic inflammatory conditions [125]. A recently published study [126] showed that resolvins, namely anti-inflammatory mediators [127,128], effectively inhibited tumor growth and enhanced the therapeutic efficacy when combined with anticancer drugs. Despite extensive studies on the role of inflammation in cancer progression, little is known regarding inflammation-independent pathways. Furthermore, the direct, non-inflammatory effects of the resulting cell necrosis on these processes have yet to be properly understood [129]. Thus far, most researchers have separately focused on the effects of either nutritional deficiency or hypoxia on angiogenesis and cancer progression [130,131,132,133,134,135,136]. Bluman et al. investigated the net total effect of necrotic cell content induced by a combination of nutrient deficiency and hypoxia on tumor cell and tumor microenvironment cues [137]. Their results showed that signals secreted by necrotic cells affected the function of various cancer cells and increased the angiogenic capacity. An analysis using high-throughput proteomic assays identified several important angiogenesis- and carcinogenesis-promoting factors that were presented or secreted by necrotic cells. Moreover, the HMGB1 antagonist carbenoxolone and the anti-VEGF bevacizumab suppressed tumor growth in combination with necrosis signaling antagonists and anticancer drugs. These results suggest that tumor cell necrosis itself facilitates an important process involved in treatment resistance during surgical resection and acts as a tissue-level cancer resistance mechanism resulting from adjuvant treatment after surgery.

### 3.2. Transfusion

Clinical studies related to blood transfusions and outcomes have shown that circumstances wherein patients receive preoperative blood products could likely affect the recurrence of cancer. Recurrence may occur depending on the preoperative nutritional and physical status, preoperative presence or absence of anemia, histology and stage of the cancer, resectability of the cancer, anesthesia procedure, volume of blood loss, stress response during the surgical procedure, and presence of complications after surgery [138,139,140,141]. The possible impact of several of the abovementioned confounding factors is complicated and difficult to deduce from retrospective studies. However, relatively few randomized trials have examined the association between blood transfusions and cancer recurrence.

According to earlier studies, most of which were retrospective in nature, allogeneic blood transfusions were associated with the increased risk of postoperative cancer recurrence and death [142,143,144,145,146,147,148,149,150,151]. Moreover, previous meta-analyses revealed an association between perioperative blood transfusions and poor prognosis after surgery for colorectal cancer and tonsil cancer [152,153]. Such findings have been supported by other retrospective studies in patients with pancreatic, liver, colorectal, prostate, and head and neck cancer [154,155,156,157,158]. In particular, a blood transfusion was found to be an independent predictor of recurrence and survival in cases of HCC and head and neck cancer. However, these results were contradicted by investigators who could not identify an association between cancer recurrence and blood transfusions [159,160,161,162,163,164,165]. The best current evidence may perhaps arise from a large meta-analysis performed by the Cochrane Group, in which pooled estimates of the effect of a perioperative blood transfusion on recurrence from randomized studies yielded an odds ratio (OR) of 1.42 (95% confidence interval (CI) = 1.20–1.67; *p* < 0.0001) for transfused patients. Although heterogeneity was observed, a stratified meta-analysis verified these findings according to the disease site and stage, timing of blood product administration, type of product administered, and transfusion product dose. However, considering the heterogeneity of samples and the inability to analyze the effect of the surgical procedure, the authors were unable to establish a clear causal relationship [166].

It remains unclear whether differences in the risk of cancer recurrence exist following allogeneic or autologous transfusions [167,168]. An observational study suggested that, as compared to an autologous blood transfusion, allogeneic transfusion in patients who underwent head and neck cancer surgery was associated with a 40% higher cancer recurrence [169]. However, this result contradicted the findings of a randomized controlled trial involving patients with colorectal cancer undergoing surgical tumor resection who were assigned to receive allogeneic or autologous blood transfusions [170]. Despite the limited data, preoperative autologous blood collection and an intraoperative autologous blood transfusion appear to be safe for patients with HCC [171,172]. In fact, a small retrospective study suggested that patients who received autologous blood had longer long-term cancer-free survival than those who received allogeneic blood [172]. Intraoperative cell salvage techniques may be used in patients who refuse major surgery or allogeneic blood products. However, one concern regarding the use of cell salvage techniques during tumor surgery has been the possibility of reinjecting malignant cells obtained from the surgical site [173,174]. For example, cancer cells have been observed in surgically collected and double-filtered blood samples from patients with HCC who underwent liver transplantation [175]. However, at least one study has shown a lack of cellular staining for malignant tumor markers after filtration [176]. Most importantly, intraoperative cell salvage techniques and autologous blood transfusions do not appear to have an empirical impact on the recurrence rates in patients undergoing cancer surgery [177,178,179].

### 3.3. Perioperative Period

Studies have suggested that short perioperative periods (defined as days to weeks after tumor resection) promote the formation of new metastases and growth or the development of existing micrometastases through numerous pro-metastasis, pro-angiogenic, and immunosuppressive mechanisms [180]. Conversely, the surgical removal of the primary tumor has also been suggested to prevent or decrease metastatic progression through various mechanisms. These include the cessation of tumor cell shedding into the circulatory and lymphatic systems; the removal of metastasis-promoting factors, such as IL-6, IL-8, and VEGF, that are secreted by the tumors; the termination of the immunosuppressive effects of resected tumors. These multiple metastasis-promoting and anti-metastatic processes occur concurrently during short perioperative periods, the cumulative effects of which may tilt the scale toward either metastatic proliferation or disease arrest or elimination. Patients without obvious metastatic disease have been considered to possess a dynamic balance (i.e., “preoperative balance”) between pro-metastatic and antimetastatic processes that may tilt slightly toward metastatic progression before surgery. In many cases, however, a storm of metastasis-promoting and -suppressing effects rages before surgery [181], tipping the scale toward the development of metastasis. Moreover, the pro-metastatic effects are activated by each other, creating a self-perpetuating “snowball effect” that promotes disease recurrence. However, perioperative therapeutic interventions targeting ≥ 1 pro-metastatic process may result in a shift toward disease cessation or elimination. Thus, the perioperative period is important in determining the risk of postoperative metastatic disease and provides a window of therapeutic scope for residual malignant disease. Regrettably, this window of opportunity has been largely undetected clinically [182].

## 4. Clinical Indices

Among the clinical indicators of preoperative inflammation that have been identified as predictive markers of postoperative outcomes, GPS, SII, and NLR are well-known and are major indicators for several types of cancer. Table 1 shows the relationship between these indicators and the clinical outcomes.

## 5. Conclusions

Although the number of clinical reports on the associations between postoperative recurrence and the inflammatory state of the tumor microenvironment varies according to cancer type, inflammation was shown to be markedly associated with cancer recurrence and progression in nearly all the major types of cancer. Further translational studies linking laboratory data and clinical practice will help to clarify the association between cancer recurrence and inflammation, as well as to establish more effective postoperative treatment and identify lifestyle habits to avoid recurrence.

## Figures and Tables

**Table 1 jcm-10-00228-t001:** Relationship between clinical indices and clinical outcomes after resection.

Clinical Indices	Cancer Types	Clinical Outcomes
Glasgow Prognostic Score	NSCLC Gastric cancer HCC Pancreatic cancer Ovarian cancer Cervical cancer Colorectal cancer Bladder cancer	Poor OS [42,183] Poor OS [184,185] Poor OS [186,187] Poor OS [95,188] Poor OS [103,104,105,189] Poor OS and DFS [109] Poor OS [190,191] Poor RFS [192]
Systemic Immune-Inflammation Index	All NSCLC Small cell lung cancer Esophageal cancer Gastric cancer Pancreatic cancer Urological cancer Melanoma	Poor OS, RFS, DFS [193] Poor OS [39,193], poor DFS [39] Poor OS [193] Poor OS [38,193] Poor OS [193,194], poor DFS [194] Poor OS [193] Poor OS [193] Poor OS [193]
Neutrophil-to-Lymphocyte Ratio	All NSCLC Gastric cancer HCC Colorecal cancer Renal cancer	Poor DFS [195] Poor RFS [34,36] Poor OS [196], poor DFS [195,196] Poor OS [88], poor DFS [87,195] Poor OS [197], poor DFS [195,197] Poor DFS [195]

NSCLC, non-small cell lung cancer; HCC, hepatocellular carcinoma; OS, overall survival; DFS, disease-free survival; RFS, recurrence-free survival.

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
