# Peer review of "Links between Inflammation and Postoperative Cancer Recurrence"

_jcm, 2021, doi:10.3390/jcm10020228_

Round 1

Reviewer 1 Report

The authors provide an interesting review on the importance of systemic markers of inflammation on tumour outcome in various tumour entities. The paper is well written and very informative. There are only a few suggestions for improvement. 

The literature is not complete. As examples, the following two citations are suggested

: Lindenmann, J; Fink-Neuboeck, N; Taucher, V; Pichler, M; Posch, F; Brcic, L; Smolle, E; Koter, S; Smolle, J; Smolle-Juettner, FM
Prediction of Postoperative Clinical Outcomes in Resected Stage I Non-Small Cell Lung Cancer Focusing on the Preoperative Glasgow Prognostic Score.
Cancers (Basel). 2020; 12(1)

Lindenmann, J; Fink-Neuboeck, N; Koesslbacher, M; Pichler, M; Stojakovic, T; Roller, RE; Maier, A; Anegg, U; Smolle, J; Smolle-Juettner, FM
The influence of elevated levels of C-reactive protein and hypoalbuminemia on survival in patients with advanced inoperable esophageal cancer undergoing palliative treatment. J Surg Oncol. 2014; 110(6):645-650

Headline "3. Inflammation-Related Disorders" is misleading. None of the following subheadings (Necrosis; Transfusion; Postoperative Period) is a "disorder". "Inflammation-Related Conditions" may be suggested als an alternative. 

The chapter "4. Clinical Indices" shows some overlap with chapter 2, but while chapter 2 is subdivided along tumour entities, chapter 4 is subdivided along specific clinical indices. It is not clear why exactly three clinical indices have been selected and, for example, the widely used Glasgow Prognostic Score is not included. It is suggested that chapter 2 remains in its narrative form, whilc chapter 4 could be replaced by a comprehensive table including all inflammatory indices mentioned in the paper. 

Author Response

Reviewer #1:

  1. The literature is not complete.

Reply 1: Thank you for your remarks on this point. We have also taken into account the suggestions of other reviewers and have added some descriptions in the manuscript.

  1. Headline "3. Inflammation-Related Disorders" is misleading. None of the following subheadings (Necrosis; Transfusion; Postoperative Period) is a "disorder". "Inflammation-Related Conditions" may be suggested also an alternative.

Reply 2: As you suggested, we have replaced the headline 3 with "Inflammation-Related Conditions".

  1. The chapter "4. Clinical Indices" shows some overlap with chapter 2, but while chapter 2 is subdivided along tumor entities, chapter 4 is subdivided along specific clinical indices. It is not clear why exactly three clinical indices have been selected and, for example, the widely used Glasgow Prognostic Score is not included. It is suggested that chapter 2 remains in its narrative form, while chapter 4 could be replaced by a comprehensive table including all inflammatory indices mentioned in the paper.

Reply 3: Thank you for your suggestion. We have shortened the description regarding clinical indices and added a comprehensive table showing inflammation-related indices in Table 1.

Reviewer 2 Report

In the present paper the Authors investigated the association between inflammation and cancer recurrence after surgical resection, focusing on inflammatory indicators and cancer recurrence predictors according to the type of cancer and clinical indicators.
The work is of sure interest for the oncologic community, it is complete and it is analyzed all involved aspects. The references are rich and include recent publications. 

I can recommend the paper for publication in the Journal of Clinical Medicine.

Author Response

Reviewer #2:

In the present paper the Authors investigated the association between inflammation and cancer recurrence after surgical resection, focusing on inflammatory indicators and cancer recurrence predictors according to the type of cancer and clinical indicators.
The work is of sure interest for the oncologic community, it is complete and it is analyzed all involved aspects. The references are rich and include recent publications. 

I can recommend the paper for publication in the Journal of Clinical Medicine.

Reply 1: Thank you very much for your kind comments on our manuscript.

Reviewer 3 Report

Thank you very much for the opportunity of reviewing this article. I congratulate the authors for the effort to review the large volume of information on the subject. However, there are so much information with little specificity of the results. In fact, the conclusions of the review do not include a synthesis of new information to that already known until now. Abstract does not include any result or conclusion.  

Other major concerns: 

- More attention should be put in selecting the most recent evidence about the theme. Just two references of 172 are articles published in 2020. For example:

  1. Oshi M, et al. Intra-Tumoral Angiogenesis Is Associated with Inflammation, Immune Reaction and Metastatic Recurrence in Breast Cancer. Int J Mol Sci. 2020;21(18):6708.
  2. Soria F, et al. Prognostic value of the systemic inflammation modified Glasgow prognostic score in patients with upper tract urothelial carcinoma (UTUC) treated with radical nephroureterectomy: Results from a large multicenter international collaboration. Urol Oncol;38(6):602.e11-602.e19.
  3. Bednarz-Misa I, et al. Distinct Local and Systemic Molecular Signatures in the Esophageal and Gastric Cancers: Possible Therapy Targets and Biomarkers for Gastric Cancer. Int J Mol Sci. 2020;21(12):4509.

- A greater discussion on the results of the chosen articles (methodology aspects, characteristics of patients, characteristics of tumour…) should have been carried out. For example, to indicate if there is heterogeneity in the combined analyses when the results of a metaanalysis are presented. This aspect is even more important when some parts of the review are focused in the results of just one article: reference 66 in esophageal cancer.

Minor revisions:

- There is a possible mistake in the references 96-99 in line 221

Author Response

-Reviewer #3:

  1. There is so much information with little specificity of the results. In fact, the conclusions of the review do not include a synthesis of new information to that already known until now. Abstract does not include any result or conclusion.

Reply 1: Thank you for your comment. Although new information may be limited in our manuscript, we focused on clinical indicators rather than molecular interactions between tumor recurrence and the microenvironment to make it easier for our readers, mainly physicians and surgeons, to understand. On the other hand, there are few review articles describing the relationship between inflammation and postoperative recurrence according to the cancer type. We selected the major cancer types and discussed the relationship between inflammation and postoperative recurrence in each type of cancer. We believe that our viewpoint is novel and provides useful insights.

  1. More attention should be put in selecting the most recent evidence about the theme.

Reply 2: I agree with your suggestion. We have cited additional recent papers, including the one you kindly detailed, and added discussion in the manuscript.

  1. A greater discussion on the results of the chosen articles (methodology aspects, characteristics of patients, characteristics of tumor…) should have been carried out. For example, to indicate if there is heterogeneity in the combined analyses when the results of a metanalysis are presented. This aspect is even more important when some parts of the review are focused in the results of just one article: reference 66 in esophageal cancer.

Reply 3: Thank you very much for your very important remarks. According to the suggestions, we have revised the descriptions in Chapter 2.

  1. There is a possible mistake in the references 96-99 in line 221.

Reply 4: Thank you. We have corrected the citation to “83-85”.

Round 2

Reviewer 1 Report

Concerning headline 3: This headline sill reads "Inflammation-Related Disorders" despite the claim of the authors that they have changed the headline as suggested into "Inflammation-Related Conditions". 

Author Response

Concerning headline 3: This headline sill reads "Inflammation-Related Disorders" despite the claim of the authors that they have changed the headline as suggested into "Inflammation-Related Conditions". 

Reply 1: As you suggested, we have replaced the headline 3 with "Inflammation-Related Conditions". Thank you for your thoughtful comments.

Reviewer 3 Report

The paper has improved and it is suitable for publication.

Author Response

The paper has improved and it is suitable for publication.

Reply 1: Thank you very much for your kind comments on our manuscript.